# Continuous extracorporeal femoral perfusion model for intravascular ultrasound, computed tomography and digital subtraction angiography

Philipp Gruschwitz[1]◉*, Viktor Hartung[1]◉, Florian Kleefeldt[2], Dominik Peter[3], Sven Lichthardt[3], Henner Huflage[1], Jan-Peter Grunz[1], Anne Marie Augustin[1], Süleyman Ergün[2], Thorsten Alexander Bley[1], Bernhard Petritsch[1]

1 Department of Diagnostic and Interventional Radiology, University Hospital of Würzburg, Würzburg, Germany, 2 Institute of Anatomy and Cell Biology, University of Würzburg, Würzburg, Germany, 3 Department of General, Visceral, Transplant, Vascular, and Pediatric Surgery, University Hospital of Würzburg, Würzburg, Germany

◉ These authors contributed equally to this work.
* gruschwitz_p@ukw.de

## Abstract

### Objectives

We developed a novel human cadaveric perfusion model with continuous extracorporeal femoral perfusion suitable for performing intra-individual comparison studies, training of interventional procedures and preclinical testing of endovascular devices. Objective of this study was to introduce the techniques and evaluate the feasibility for realistic computed tomography angiography (CTA), digital subtraction angiography (DSA) including vascular interventions, and intravascular ultrasound (IVUS).

### Methods

The establishment of the extracorporeal perfusion was attempted using one formalin-fixed and five fresh-frozen human cadavers. In all specimens, the common femoral and popliteal arteries were prepared, introducer sheaths inserted, and perfusion established by a peristaltic pump. Subsequently, we performed CTA and bilateral DSA in five cadavers and IVUS on both legs of four donors. Examination time without unintentional interruption was measured both with and without non-contrast planning CT. Percutaneous transluminal angioplasty and stenting was performed by two interventional radiologists on nine extremities (five donors) using a broad spectrum of different intravascular devices.

### Results

The perfusion of the upper leg arteries was successfully established in all fresh-frozen but not in the formalin-fixed cadaver. The experimental setup generated a stable circulation in each procedure (ten upper legs) for a period of more than six hours. Images acquired with CT, DSA and IVUS offered a realistic impression and enabled the sufficient visualization of

**Data Availability Statement:** All relevant data are within the paper and its Supporting Information files.

**Funding:** PG; Z-02CSP/18; Interdisciplinary Center for Clinical Research (IZKF) at the University of Würzburg; https://www.med.uni-wuerzburg.de/izkf/startseite/ The funders had no role in study design, data collection and analysis, decision to publish, or preparation of the manuscript. BP, TAB and JPG received speaker honoraria from Siemens Healthcare GmbH outside of the presented work. The Department of Diagnostic and Interventional Radiology of the University Hospital Würzburg receives ongoing research funding from Siemens Healthcare GmbH; https://www.siemens-healthineers.com/en-us. This publication was supported by the Open Access Publication Fund of the University of Wuerzburg. The funders had no role in study design, data collection and analysis, decision to publish, or preparation of the manuscript.

**Competing interests:** The authors have declared that no competing interests exist.

all examined vessel segments. Arterial cannulating, percutaneous transluminal angioplasty as well as stent deployment were feasible in a way that is comparable to a vascular intervention in vivo. The perfusion model allowed for introduction and testing of previously not used devices.

## Conclusions

The continuous femoral perfusion model can be established with moderate effort, works stable, and is utilizable for medical imaging of the peripheral arterial system using CTA, DSA and IVUS. Therefore, it appears suitable for research studies, developing skills in interventional procedures and testing of new or unfamiliar vascular devices.

## Introduction

Intra-individual comparative studies using ionizing radiation on humans are faced with several problems. In most situations, repeated examinations are not justifiable due to medical as well as ethical concerns. Therefore, alternative investigation methods are desirable. Artificial phantoms or mock models, either commercially available or self-made, allow for repeated examinations but often constitute an oversimplification of anatomy and consequently introduce methodological limitations. Among the issues worth mentioning in particular are an unrealistically low attenuation of radiation, artifacts caused by artificial materials used, and impracticable or unrealistic interventional procedures for various reasons such as complexity, depiction of pathologies or unrealistic haptic feedback [1–3]. Animal models facilitate in-vivo examinations but require appropriate framework conditions and are ethically debatable [4]. Because of their individual limitations, both model types complicate the transformation process to clinical practice, comparability, and usability for realistic training/teaching. Human cadaveric perfusion models have been reported as a potential alternative, however, most established models have been developed for training in vascular surgery with perfusion of the central arterial system [5–11]. Lately, a static model of the peripheral arterial system using dissected lower extremities has been suggested [12], but a dynamic perfusion model of the peripheral arterial system with continuous circulation is still missing.

To address this research gap, we present a novel femoral perfusion model based on human body donors, which allows for examinations of the upper leg using computed tomography angiography (CTA), intravascular ultrasound (IVUS) and digital subtraction angiography (DSA), while also facilitating realistic interventional procedures. The aims of this study were to introduce the concept of a continuous extracorporeal femoral perfusion model, to validate its feasibility and to evaluate the model regarding its usability for research, teaching, training, and device testing.

## Materials and methods

### Human cadavers

To develop the perfusion model, we used one female formalin-fixed and five fresh-frozen adult whole body human cadavers (three female and two male). All specimens were obtained from the local anatomical institute in accordance with national and European law [13]. In their lifetime, donors had authorized the usage of their bodies for education and research purposes. Additionally, permission for this experimental study was granted by the local

institutional review board. In accordance with our institution's policy, demographics and medical details of the donors remain classified.

Fresh-frozen cadavers were iced post-mortem (< 24 hours after death) at a temperature of -20˚C without additional chemical conservation for a short time of several weeks to a few months. The defrosting process was started 48–72 hours before study commencement depending on the body mass of the body donors. Therefore, bodies were stored in a refrigerated room at about 7˚C to achieve a final core temperature of 3–5˚C. The formalin-fixed cadaver was perfused via a femoral access with a 10-fold diluted buffered formalin solution (~4% formaldehyde) for about 24 hours and stored in a formalin-filled tank for at least 12 months.

## Perfusion model

Exact product descriptions and manufacturer specifications of the applied materials are summarized in Table 1. For the denomination of the applied materials, we report roman numerals in parenthesis.

**Preparation of the vascular accesses.** A non-contrast CT scan to obtain an initial overview of the vascular anatomy was performed in three of five cases prior to the cadaver preparation.

First, the infragenicular popliteal artery (APOP) was prepared and dissected as distally as possible (Fig 1A) using a medial access between Mm. gracilis/semitendinosus and M. gastrocnemius. Mechanic thrombectomy using a Fogarty embolectomy catheter (I) was found to be useful to remove the major burden of intravascular clots (Fig 1B). To verify vessel patency and as a guide rail for the introducer sheaths, it was helpful to insert a stiff, hydrophilic-coated 0.035" guidewire with angled flexible tip (II). For the wire to be advanced without image guidance, we suggest to perform this procedure in reversed flow direction from distal to proximal, as this will reliably avert diversion of the wire in side branches. Second, the external iliac artery (AIE) and common femoral artery (AFC) were prepared after a vertical inguinal skin incision (Fig 1C). After subsequent ligature of the distal AIE right above the inguinal ligament and the

**Table 1. Summary of the product descriptions and manufacturer specifications of the applied materials.**

| Number | Product Designation | Manufacturer | Manufacturer Origin |
|---|---|---|---|
| I | Synthel® Silicone—Regular Tip | LeMaitre Vascular Inc. | Burlington, Massachusetts, USA |
| II | Radifocus® GuideWire M 180 cm | Terumo Corporation | Tokyo, Japan |
| III | Radifocus® Introducer II 10 cm | Terumo Corporation | Tokyo, Japan |
| IV | Cable Strap BN 2.5 x 100 mm | fischerwerke GmbH & Co. KG | Waldachtal, Germany |
| V | SUPOLENE, Resorba® | Advanced Medical Solutions Group plc | Winsford, United Kingdom |
| VI | Double Head Peristaltic Pump G728-2 | Shenzhen Grothen Technology Co. Ltd | Shenzhen, China |
| VII | Heidelberger Extension 75 cm AMT0034 | AMT Medical GmbH | Holzerlingen, Germany |
| VIII | Stopcock Discofix-C® | B. Braun SE | Melsungen, Germany |
| XI | Secretion Bag lock-fitting for Pleurofix® | B. Braun SE | Melsungen, Germany |
| X | Urine Bag 1500 ml CH 15 | P.J. Dahlhausen & Co. GmbH | Köln, Germany |
| XI | Connector funnel CH 15/Luer-Lock | B. Braun SE | Melsungen, Germany |
| XII | Enema Bag 2000 ml CH 15 | Primed Halberstadt Medizintechnik GmbH | Halberstadt, Germany |
| XIII | Ringer's Solution 1000 ml Plastipur® | Fresenius Kabi AG | Bad Homburg, Germany |
| XIV | Glucose-20% Solution 1000 ml | B. Braun SE | Melsungen, Germany |
| XV | Male/male Luer-Lock Connector | B. Braun SE | Melsungen, Germany |
| XVI | Peritrast® 400mg/ml iodine | Dr. Franz Köhler Chemie GmbH | Bensheim, Germany |
| XVII | Contrast Media Injector CT motion™ | ulrich GmbH & Co. KG | Ulm, Germany |
| XVIII | Pattex® Super Glue Liquid | Henkel AG | Düsseldorf, Germany |

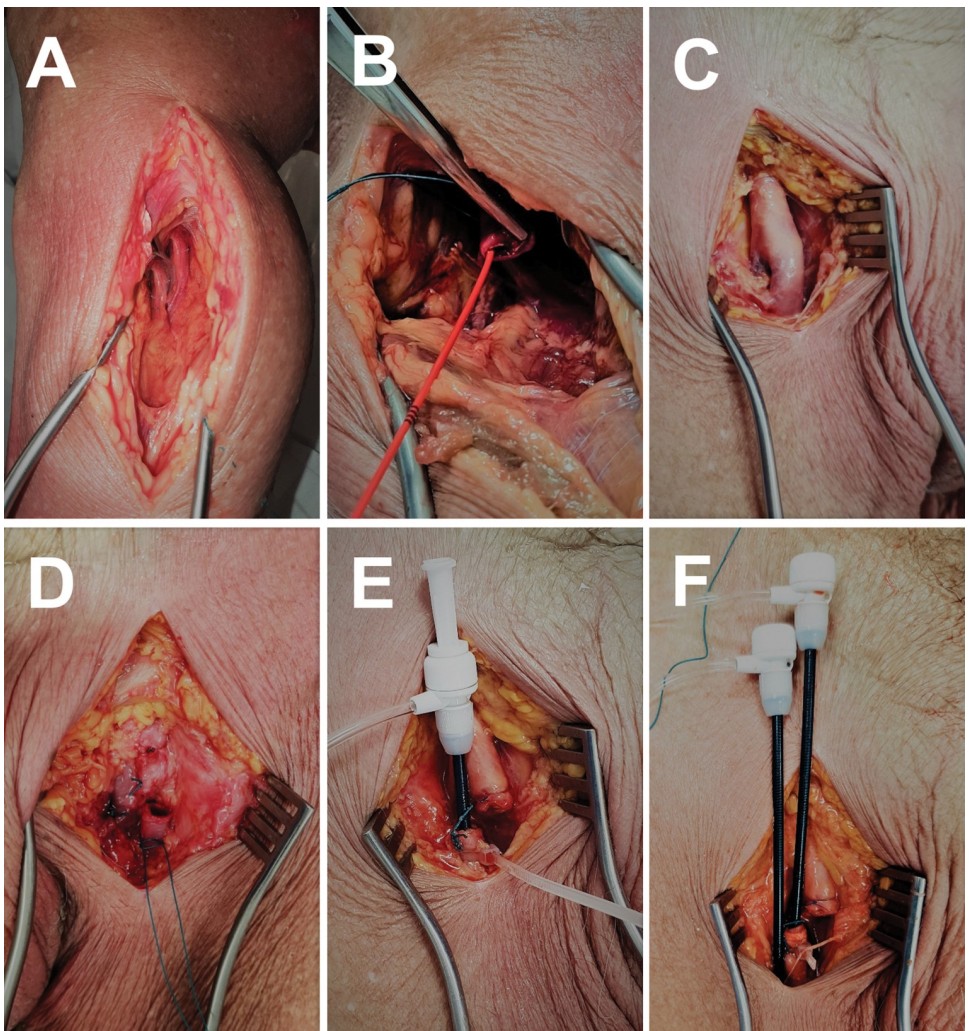

**Fig 1. Surgical access.** A. Situs after preparation of the infragenicular access to the distal popliteal artery using a medial access between Mm. gracilis/semitendinosus and M. gastrocnemius. B. Infragenicular outflow with introduced Fogarty embolectomy catheter in popliteal artery. C. Inguinal situs after preparation of the femoral bifurcation. D. Severed common femoral artery after proximal ligature of the common femoral artery and deep femoral artery. E. Introducer sheath placed in the stump of the common femoral artery and fixation by knotting and cable strap. F. Inguinal situs after insertion of the working introducer sheath in the distal common femoral artery with standard Seldinger technique.

deep femoral artery, an incision was made in the AFC distal to the ligature (Fig 1D). Then a 7-French reinforced introducer sheath (III) was positioned over the wire into the vessel and fastened by knotting a small cable tie (IV). For this, we recommend using a synthetic, non-absorbable, braided thread with large diameter, approximately USP 0 (metric 3.5) (V). The resulting situs is shown in Fig 1E. Afterwards, the superficial femoral artery (AFS) and APOP were rinsed thoroughly using the introducer sheath to remove residual thrombotic material and detritus to avoid subsequent blockage. Subsequently, another 7-French introducer sheath was inserted over the wire in the stump of the APOP and fixed as previously mentioned to create an outflow valve. In preparation of the interventional procedures, a third 7-French introducer sheath was placed parallel to the influx sheath acting as working sheath. This was done via Seldinger technique after a working circulation had been established with the puncture site located just distally of the inflow sheath.

## Establishment of the extracorporeal perfusion

A double-head peristaltic pump (VI) generated the necessary perfusion pressure. Using a double-headed pump allowed to establish a closed circulation with two single circuits; one femoral perfusion circuit and one reservoir or rather drain circuit to fill up any lost amount of fluid or empty the system for e.g., altering the contrast media dosage or injecting a contrast agent bolus during DSA. Schematic illustration of the experimental setup is given in Fig 2.

In order to establish the extracorporeal perfusion, the reservoir/drain circuit needs to be set up first. To achieve this, the male connector of an extension line type Heidelberger with 3 mm inner diameter (VII) was cut and the open end was connected to the build-in tube of pump #2 on the efflux side. Then the female end of another extension line was cut and connected to the influx build-in tube of pump #1. Two three-way stopcocks (VIII) were interconnected in series by screwing the rotating male adapter to the female connector of the other cock. Alternatively, a standard three-gang manifold can be used. The stopcocks were placed between the efflux tube of pump #2 and the influx tube of pump #1. A drainage bag (IX) or urine bag (X) was attached via Luer-lock connection and a funnel-to-lock adapter (XI) to the first stopcock (efflux pump #2) as a refuse container. Furthermore, a soft enema bag (XII) with funnel-to-lock adaptor was connected to the second stopcock as a reservoir. As a flushing solution, 1 liter lactated Ringer's solution (XIII) was mixed with 1 liter of 20% glucose solution (XIV).

To establish the perfusion circuit, first, the outflow-tube of pump head #1 was connected to the influx introducer sheath located in the AFC. Second, the inflow-tube of pump head #2 was attached to the outflow introducer sheath located in the APOP. For this, the build-in tubes were attached to the open end of a Heidelberger extension line after cutting off the female connector. We suggest to use either a long extension line (1.5 m) or several interconnected short lines to bridge the distance from the pump to the introduction sheaths. Third, the male end of the extension line was connected to the straight female connector of the introducer sheath.

Subsequently, the three-way stopcock #1 (draining) was turned to straight flow direction with closed draining aperture. Stopcock #2 was brought in "T-position" to deaerate the system. For successful deaeration, the perfusion fluid bag should be lifted as high as possible to increase the hydrostatic pressure. Then, the extension tube was disconnected from the outflow introducer sheath. If the system did not deaerate sufficiently, pump #1 was started with a slow flow rate until fluid poured out of the efflux introducer sheath. Pump #2 was also turned on after the tube was reconnected. Stopcock #1 was temporarily switched to "L-position" and after complete deaeration, turned back to its initial straight position.

## CT angiography examinations

**CT technique and scan protocol.**   CTA was performed using a photon-counting CT scanner (Naeotom Alpha, Siemens Heathineers AG, Munich, Germany). For optimal image quality, we used a maximum dose protocol in "Quantum plus" ultra-high-resolution scan mode applying 120 kVp, 889.6 ± 135.9 effective mAs, 0.2 pitch and rotation time of 1 second, resulting in a computed tomography dose index of 71.2 ± 11.0 mGy.

**Continuous intraluminal contrast.**   To employ the human cadaveric perfusion model for CT examinations, another enema bag was filled with 2 liters of the flushing solution mixed with iodine contrast media. Oral contrast agents, which usually have a lower purchase price, can also be used for this purpose (XVI). The optimal mixing ratio depends on the desired contrast attenuation. Stopcock #2 was closed and attached the contrast fluid filled enema bag. Subsequently, stopcock #1 was switched to draining position and both pump heads were started to empty the system from water and wash in the contrast fluid. After approximately 2 minutes,

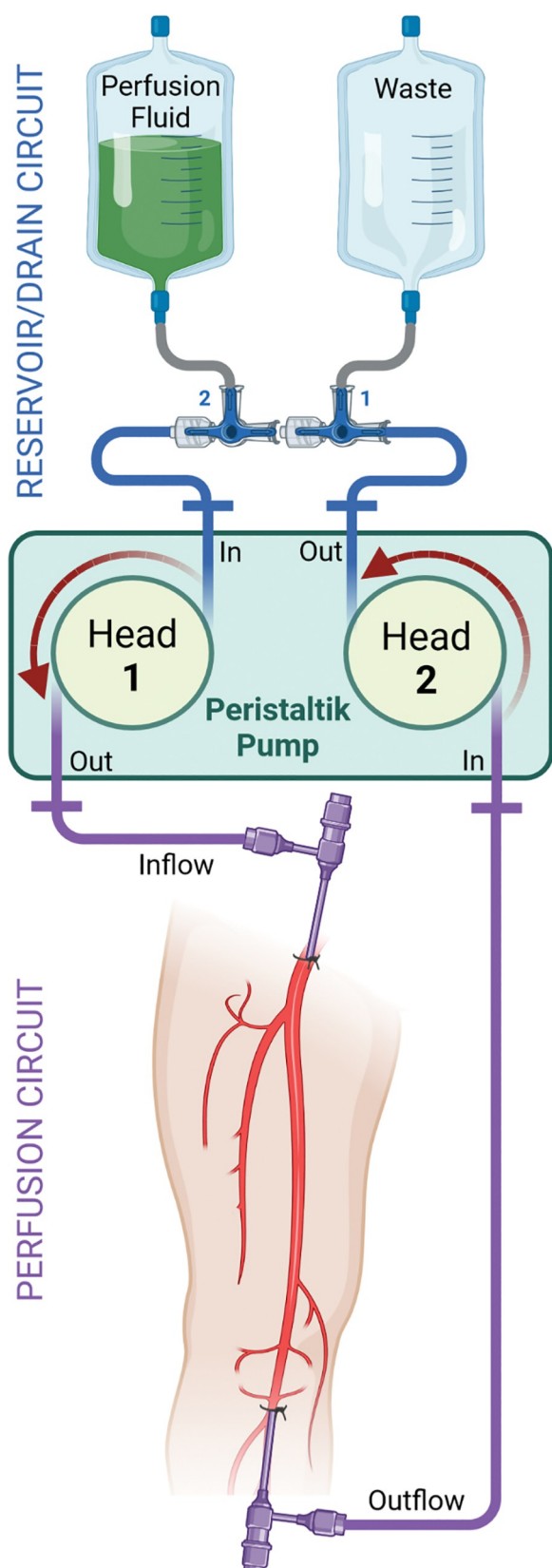

**Fig 2. Schematic illustration of the extracorporeal femoral perfusion setup.**

stopcock #1 was switched to straight position to interrupt the drainage. Hereinafter, CTA scans with continuous contrast flow are feasible using different scan protocols and scanners for comparative studies.

To perform CTA scans of both upper legs at the same time, it may be useful to connect inflow and outflow of each side to the peristaltic pump. Therefore, the side aperture of a three-way stopcock was attached to the efflux tube of pump head #1 and the influx tube of pump head #2. The male apertures of the stopcocks were connected to the introducer sheaths as mentioned previously. Notably, for connection of the female apertures, a male/male Luer-lock connector (XV) is needed.

### Bolus of contrast fluid

Alternatively, the use of a contrast fluid bolus, as applied in a clinical patient examinations, is possible. Therefore, a three-way stopcock must be interposed between the inflow tube (efflux pump head #1) and the inflow introducer sheath located in the AFC. The side aperture can then be connected to a commercially available CT contrast media injector (XVII) using the conventional patient tubing. In this scenario, circulation is established like mentioned above using saline or Ringer's solution without contrast media as perfusion fluid. Iodine contrast agent is applied as 100% contrast media bolus or mixed bolus while draining the circuit (stopcock #1). CT scans should be performed with shortest acquisition delay possible.

### Digital subtraction angiography examinations

**Additional preparation.** The extracorporeal femoral perfusion model could also be used for DSA examinations. To achieve this, a continuous perfusion with non-contrast fluid was established like described above. An additional introducer sheath for interventional procedures was inserted surgically or in Seldinger technique (Fig 1F). In either case, it is recommended to fix the introducer sheath firmly by sutures or cyanoacrylate super glue (XVIII) to avoid dislocation. To perform DSA, the flow rate of both pump heads was set to ½ of the maximum flow (~140 ml/min). Contrast agent was injected in pure or diluted fashion depending on the desired attenuation.

**DSA technique.** DSA examinations and interventions were performed using a commercially available flat detector unit (Azurion 7 C20, Philips Healthcare, Best, The Netherlands) with vendor-recommended settings for low-dose examinations of the upper leg arteries. We acquired transfemoral antegrade fluoroscopic series in posterior-anterior and 20° left/right anterior-oblique orientations with a frame rate of 1–2 images/second using the stepping-table technique after manual injection of diluted contrast medium.

### Intravascular interventions

Intravascular interventions were performed on nine extremities of five body donors by two interventional radiologists with 5 and 2 years of experience in the field. After acquisition of diagnostic series as described above, the radiologists reviewed all images, planned interventions, and discussed material use as usual. A variety of percutaneous angioplasty balloons (Pacific Plus and Medtronic Admiral Xtreme; Medtronic; Minneapolis, MN, USA), microsurgical dilatation devices (Peripheral Cutting Balloon; Boston Scientific; San Jose, CA, USA), self-expanding and balloon-mounted stents (Absolute Pro, VisiPro, and Supera; Abbott; Chicago, IL, USA), a multiple stent system for spot stenting (multi-LOC; B.Braun SE; Melsungen, Germany), as well as an endovascular dissection repair system (Tack Endovascular System; Philips Healthcare, Best; The Netherlands) were applied.

## Intravascular ultrasound

Both upper leg arteries of four donors (eigth extremities) were examined with IVUS by the two interventional radiologists. For that purpose, a commercially available intravascular ultrasound catheter with plug-and-play function was used (Philips Reconnaissance PV 018, 5 Fr, 20 MHz, Philips Healthcare, Best, The Netherlands). After thorough vessel assessment and deliberate review of CT and DSA images, sections of interest including stenoses and plaques were correlated. After angiographic stent placement, additional IVUS images were acquired and correlated with the DSA results.

# Results

## Preparation of the vascular accesses

Formalin-fixed body donors proved to be not suitable for the intends and purposes of this study due to high rigidity of the fixated soft tissue (one cadaveric specimen). In contrast, establishment of extracorporeal perfusion as described above was feasible in each upper leg of all fresh-frozen human cadavers (five cadaveric specimens). The flexibility, operability, and resistance to manipulation of vascular structures of defrosted fresh-frozen body donors were found to be very similar to living subjects/ patients. In each cadaver, the circulation was stable for at least 6 hours without unintentional interruptions.

Non-contrast CT scans prior to preparation to identify potential obstacles, such as high-grade calcifications/stenoses near the planned vascular access or occlusions of the arterial runoff, were performed before commencing the cadaver preparation in three of five body donors. Vascular access was prepared by two board-certified vascular surgeons. Together with the establishment of extracorporeal perfusion of one upper leg and the preprocedural CT scan, this step took $52 \pm 9$ min per extremity (six extremities). Without CT, the time required was $130 \pm 0$ min per extremity (two extremities). Therefore, a preprocedural CT scan shortened the preparation time by 60% (p = 0.036; unilateral Mann-Whitney U-Test). In particular, pronounced calcifications in the inguinal access and high-grade calcifications in the course of the vessel made preparation and establishment of perfusion complicated. Problems were mainly dissections and resulting subintimal material position, perforation of the dorsal vessel wall and stuck guidewires. By preceding non-contrast CT the probability of such complications could be estimated and avoided accordingly.

## Angiography examinations

Our experiences have shown that the use of a cooled hyperosmolar solution reduces the leakage and edema formation compared to physiological saline solution or tap water. Therefore, hyperosmolar solution is advisable to avoid fluid diffusion in the surrounding soft tissue with resulting edema. We employed a 50:50 mixture of Ringer's solution and 20% glucose solution resulting an osmolality of 400 mOsm/kg. Furthermore, we recommend cooling down the fluid to 5–9˚C in order to inhibit the warming of the cadaver during procedures.

To evaluate the feasibility of CTA, DSA and IVUS we subdivided the upper leg runoff in four arterial segments: proximal/middle/distal superficial femoral artery and popliteal artery (segments P1 and P2).

**Computed tomography angiography.** Extracorporeal perfusion using a continuous contrast media flow led to a stable intraluminal contrast with physiological attenuation (Fig 3A–3C). Acquired CTA scans were of accustomed appearance and almost indistinguishable from clinical scans except for some remaining intraluminal air bubbles (Fig 3D). An amount of at least 1 liter mixed contrast fluid was prepared for each scan phase to avoid the system running dry.

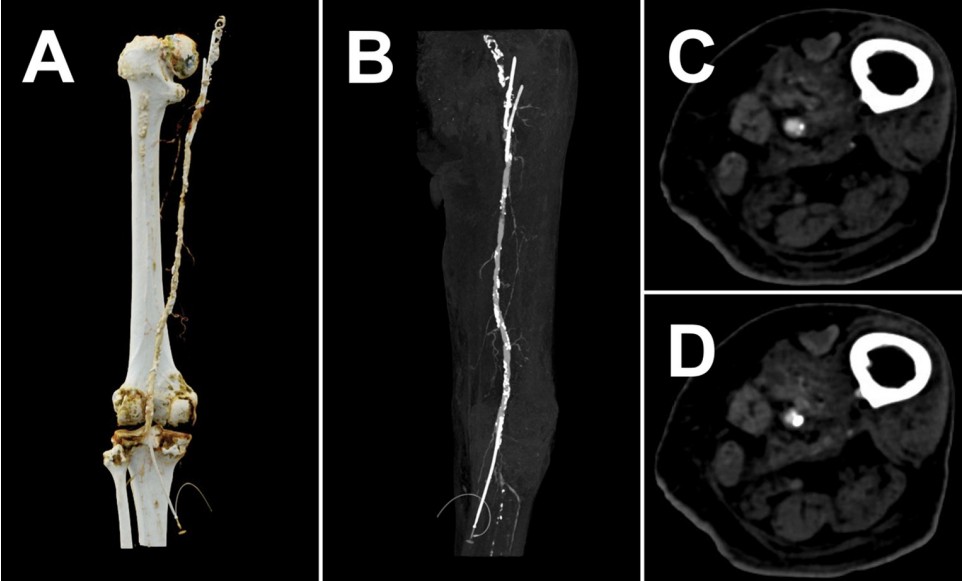

**Fig 3. Representative CTA images.** A. Three-dimensional image using virtual cinematic rendering technique (view from posterior) with partially captured proximal introducer sheaths and entirely visible distal outflow introducer sheath. B. Coronal maximum intension projection (view from anterior) of the contrasted superficial femoral and popliteal artery with arterial plaques. In addition, calcifications of the not-contrasted common femoral artery, ligated deep femoral artery, and proximal lower leg arteries are visible. C. Axial CTA slice showing a mixed plaque in the superficial femoral artery. D. Axial CTA slice with artifact by intraluminal air bubble.

**Digital subtraction angiography.** The handling of the introducer sheath does not differ from angiography examinations in patients. Using a flow rate of ½ maximum *(~140 ml/min)* simulated a high-normal cardiac output and low-normal blood pressure. Hence, direct puncture of the distal AFC resulted in a realistic backflow and introduction of the introducer sheath was possible in standard Seldinger technique. Alternatively, the vascular access could also be established surgically.

Digital fluoroscopy series were of normal appearance with realistic flush and outflow of the applied contrast fluid. In addition to the main arterial branches, even muscular branches of the upper leg arteries were visible (Fig 4).

**Intravascular interventions.** Arterial sounding and visualization with DSA were feasible in all 40 arterial segments. Transluminal balloon angioplasty (12 arterial segments) and stent placement (27 arterial segments) were practically feasible and effectively indistinguishable from common practice in patient treatment. Exemplary fluoroscopy frames are shown in Fig 5.

**Intravascular ultrasound.** IVUS was feasible in all 32 arterial segments of four body donors. Each lesion of interest in CTA and DSA was reproducible and analyzable. Even already stented vessel segments were reached by the catheter and could be analyzed. Resulting images were similar to studies in patients and of very good subjective quality (Fig 6). Doppler usage was not feasible due to the lack of particulate constituents in the perfusion solution.

## Discussion

With this study, we show that establishment of a continuous extracorporeal perfusion model is feasible in fresh-frozen body donors with acceptable financial, personal and time expenditure. The cadaveric model allows for CTA, DSA and IVUS examinations close to clinical routine with high subjective image quality, even allowing for intravascular interventions.

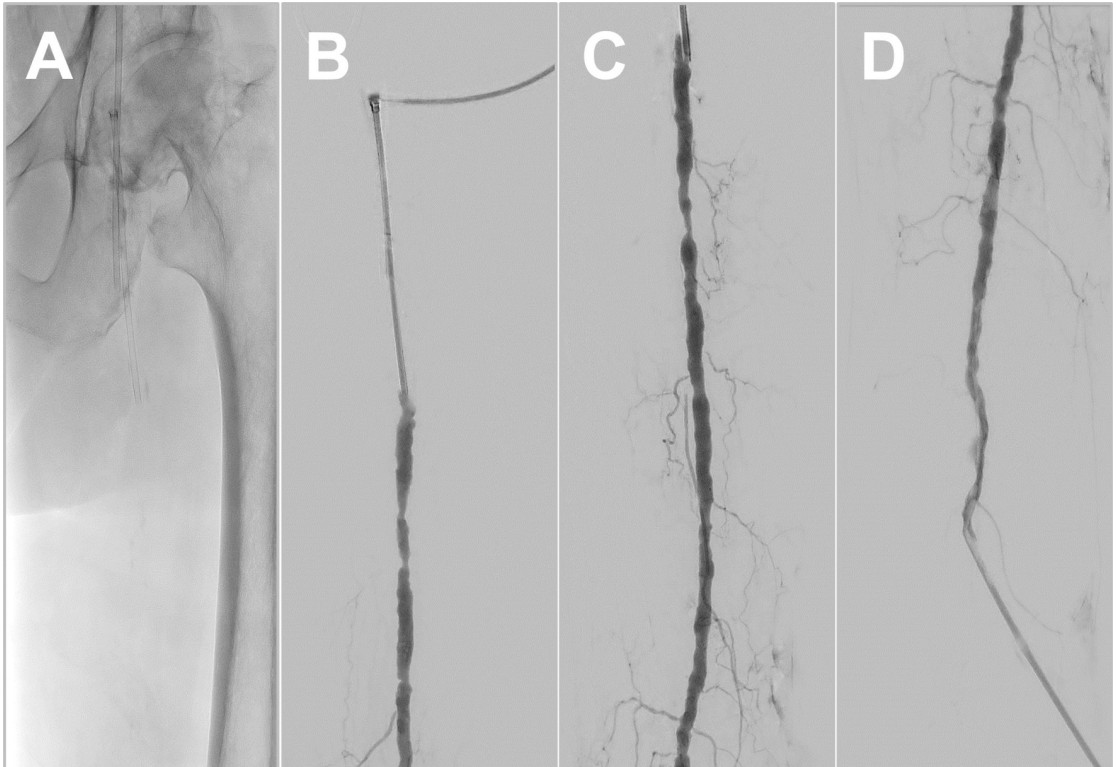

**Fig 4. Representative DSA images.** A. Fluoroscopy of the pelvis and left proximal femur with visible inflow and working introducer sheaths. B. DSA of the left proximal superficial femoral artery and contrasted working introducer sheath. C. Contrasted superficial femoral artery and muscle branches. D. DSA of the left proximal superficial femoral artery and contrasted outflow introducer sheath.

In vivo human cadaveric perfusion models with extracorporeal circulation have been suggested previously for central perfusion of the thoracic and abdominal aorta. These models were mainly used for performance testing of new technical devises or surgical methods [6, 7, 12, 14, 15] and teaching/training of medical staff [5–8, 10–12, 16] but considerably less often for research purposes [12, 16, 17]. Recently, a peripheral cadaveric model was investigated using dissected lower extremities filled up with contrast fluid, however, the lack of a simulated circulation imposed considerable restrictions [12]. While still underrepresented in the medical literature, cadaver models of the peripheral vascular system are of great interest when conducting comparative studies using ionizing radiation as well as for training of interventional procedures in protected conditions.

The presented cadaveric perfusion model offers the opportunity to test new or unfamiliar interventional devices similarly to the previously mentioned central perfusion models. Additionally, the model allows for the training of medical staff in device handling and dealing with complications. Further advantages include lifelike conditions in genuine interventional labs, representative anatomy, and pathophysiology as well as realistic tactile feedback. In contrast to the previously mentioned model using dissected lower extremities, our model offers the option to perform realistic DSA because of the continuous flow [12]. Therefore, our model can be used to simulate an entire angiographic procedure including the arterial puncture, introduction of a working sheath in Seldinger technique, diagnostic imaging, advancing guide wires and catheters, applying intravascular procedures, and completing the procedure with retraction of all devices and closure of the vascular access.

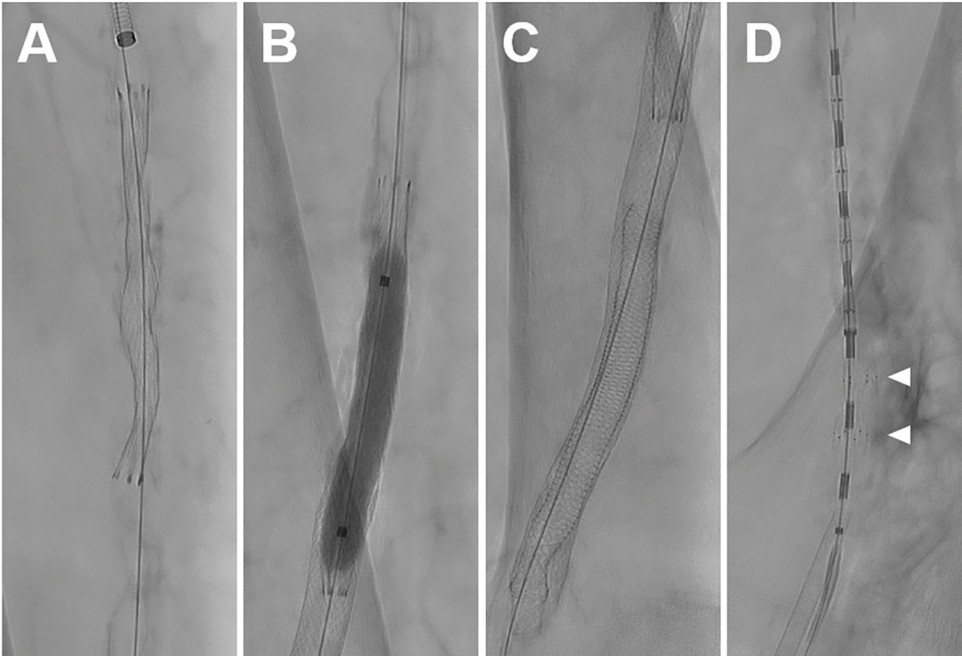

**Fig 5. Representative fluoroscopy frames of vascular interventions.** A. Fluoroscopy image of the right upper leg with under-expansion of a self-expanding-stent inside the stenotic superficial femoral artery. B. Inflated PTA balloon inside a stented vessel segment for remodeling. C. Fluoroscopy frame of a stent-in-stent situation located in the distal right superficial femoral artery. D. Fluoroscopy frame of an endovascular dissection repair device (Tack Endovascular System, Philips Healthcare) with two expanded tacks (marked with white arrowheads) and four still mounted on the multi-delivery system.

Furthermore, the introduced model allows for repeated examinations of the same specimen with ionizing radiation and invasive procedures which is not feasible in living individuals. This offers the option to investigate advantages of novel imaging techniques as well as comparison with reference standards. Moreover, it enables to use the best possible reference, e.g. maximum dose CT scans, invasive reference standards like IVUS or even histological correlation.

The establishment of the perfusion model revealed some pitfalls. Formalin-fixed body donors offer simplified handling regarding cooling and conservation but were not

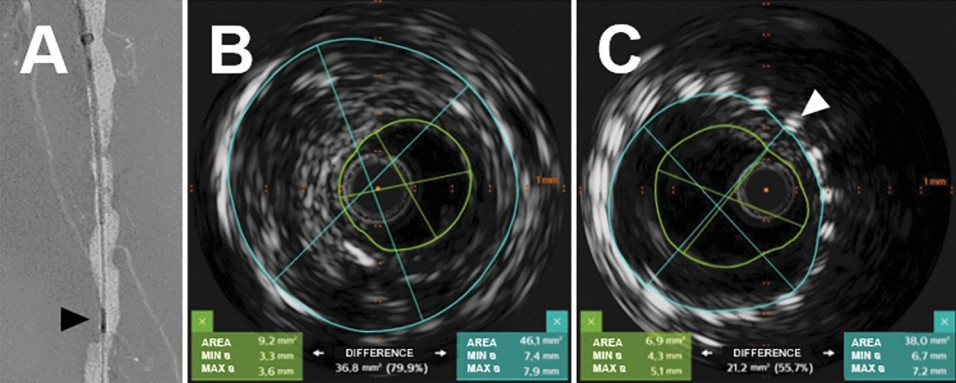

**Fig 6. Representative IVUS images.** A. DSA showing the IVUS catheter in intravascular position. B. IVUS image of an arterial segment with stenosis of ~80%. Imaging plane is marked with a black arrowhead in (A). (blue line: suspected vessel diameter; green line: residual perfused lumen). C. IVUS image generated in a stented vessel segment (metal reflex marked with a white arrowhead) with a low-grade in stent stenosis.

advantageous in a lot of other respects, such as rigidity of soft tissue and vascular structures which made preparation difficult/impossible. Furthermore, the fixation process leads to undesirable and unforeseeable tissue changes which could influence the examination results. Arbatli et al. [15] reported that intravascular thrombotic clots are difficult to remove in formalin-fixed body donors and suggested the use of fresh-frozen cadavers, in which clots can easily be removed by flushing the vessels with saline solution or mechanical extraction using a Fogarty catheter. Moreover, vessel structures in fresh-frozen cadavers behaved similarly to vessels in patients in all aspects under study. Nevertheless, we recommend involving vascular surgeons in the preparation process and preceding it with a non-contrast CT to detect calcifications, positional abnormalities, as well as similar variations to avoid resulting problems with the establishment of perfusion.

According to Arbatli et al. [15], the addition of glycerin or other unphysiological substances to increase the negative osmotic pressure gradient of the perfusion fluid is not necessary. We used Ringer's solution and 20% glucose solution in equal parts to produce a hyperosmolar solution with refrigerator temperature to decelerate the warming process. Low flow rates allowed for longer intervention procedures. However, it should be noted that capillary leakage and tissue edema increases continuously during the examination and may impair analyzability by iodine accumulation over time, especially in CT diagnostics. For this reason, CTA examinations are more time-sensitive than DSA, IVUS, and vascular procedures. This should be accounted for in the experimental design and we recommend support by vascular surgeons for expeditious and complication-free preparation.

We acknowledge that some limitations are associated with the presented perfusion model. First, we established the extracorporeal perfusion in only five body donors limiting generalizability of results. Presumably, high-grade arterial calcifications significantly complicate the establishment of an extracorporeal perfusion. Second, additional vascular interventions and procedures have to be tested bevor a general recommendation for training and device testing with the proposed model can be given. Third, the model may be transferable to the upper extremity but is probably not suitable for central perfusion simulation as this would require more elaborate preparation of vascular access and circulation. Last, cadavers could be used just a single time for this model.

## Conclusion

The presented extracorporeal femoral perfusion model can be reliably established in fresh frozen body donors with moderate expenditure in resources. It enables a stable perfusion of the upper leg arteries and facilitates realistic examination conditions for CTA, DSA, IVUS, and vascular interventions. The model provides realistic image appearances as well as mechanical and practical properties, hence allowing for training of interventional procedures and device testing. Further studies are necessary to explore the full potential and opportunities of the proposed model.

## Supporting information

**S1 File.**
(PDF)

## Acknowledgments

The authors would like to thank the staff or the Institute of Anatomy and Cell Biology of the University of Würzburg and the Department of Diagnostic and Interventional Radiology of

the University Hospital Würzburg not listed as authors for their support through the development of the experimental perfusion model.

## Author Contributions

**Conceptualization:** Philipp Gruschwitz, Viktor Hartung, Dominik Peter, Sven Lichthardt, Henner Huflage, Jan-Peter Grunz, Süleyman Ergün, Thorsten Alexander Bley, Bernhard Petritsch.

**Data curation:** Philipp Gruschwitz, Viktor Hartung, Anne Marie Augustin.

**Formal analysis:** Philipp Gruschwitz.

**Funding acquisition:** Philipp Gruschwitz, Jan-Peter Grunz, Thorsten Alexander Bley, Bernhard Petritsch.

**Investigation:** Philipp Gruschwitz, Viktor Hartung, Florian Kleefeldt, Dominik Peter, Sven Lichthardt, Henner Huflage, Jan-Peter Grunz, Anne Marie Augustin.

**Methodology:** Philipp Gruschwitz, Viktor Hartung, Florian Kleefeldt, Dominik Peter, Sven Lichthardt, Henner Huflage, Jan-Peter Grunz, Anne Marie Augustin, Bernhard Petritsch.

**Project administration:** Philipp Gruschwitz.

**Resources:** Philipp Gruschwitz, Viktor Hartung, Florian Kleefeldt, Süleyman Ergün.

**Software:** Philipp Gruschwitz.

**Supervision:** Viktor Hartung, Jan-Peter Grunz, Thorsten Alexander Bley, Bernhard Petritsch.

**Validation:** Philipp Gruschwitz, Viktor Hartung.

**Visualization:** Philipp Gruschwitz, Henner Huflage.

**Writing – original draft:** Philipp Gruschwitz, Viktor Hartung.

**Writing – review & editing:** Philipp Gruschwitz, Henner Huflage, Jan-Peter Grunz, Anne Marie Augustin, Bernhard Petritsch.

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
