## [Decision Letter · Decision Letter 0]

28 Mar 2023

PONE-D-23-03751Continuous extracorporeal femoral perfusion model for intravascular ultrasound, computed tomography and digital subtraction angiographyPLOS ONE

Dear Dr. Gruschwitz,

Thank you for submitting your manuscript to PLOS ONE. After careful consideration, we feel that it has merit but does not fully meet PLOS ONE’s publication criteria as it currently stands. Therefore, we invite you to submit a revised version of the manuscript that addresses the points raised during the review process.

We look forward to receiving your revised manuscript.

Kind regards,

Peter R. Corridon

Academic Editor

PLOS ONE

Journal Requirements:

Reviewers' comments:

Reviewer's Responses to Questions

**Comments to the Author**

1. Is the manuscript technically sound, and do the data support the conclusions?

Reviewer #1: Yes

Reviewer #2: Yes

2. Has the statistical analysis been performed appropriately and rigorously? 

Reviewer #1: N/A

Reviewer #2: N/A

3. Have the authors made all data underlying the findings in their manuscript fully available?

Reviewer #1: Yes

Reviewer #2: Yes

4. Is the manuscript presented in an intelligible fashion and written in standard English?

Reviewer #1: Yes

Reviewer #2: Yes

5. Review Comments to the Author

Reviewer #1: The authors have developed an extracorporeal perfusion model for the femoral artery in human cadavers. This model is of interest for research and training purposes, especially in the fields of radiology and vascular surgery. Therefore, the model may also contribute to the reduction of the use of large animals such as pigs, which are sometimes used for training purposes.

The authors have described setting up the model in great detail. This will certainly help anatomists and clinicians to set up a similar model and experiment with other applications as well.

The paper is well written and well discussed including its potential applications and limitations.

I have the following comments.

1. the authors have decided to use the common clinical terms for the arteries instead of the official anatomical terminology. This is fine but I suggest that they:

- explain the choice for terminology. In Terminologia Anatomica: femoral artery gives off the deep artery of thigh (as a branch). Clinical terminology: common femoral artery divides into superficial and deep femoral arteries.

- replace “profound femoral artery” by “deep femoral artery”

2. M&M, lines 37-39. Formalin is a solution of 37% formaldehyde. It is common to use a 4% formaldehyde solution for fixation purposes. This is approximately 10% formalin. It seems to me that 10% formaldehyde, as the authors state, is not correct.

3. M&M, line 50. The infragenicular approach of the popliteal artery is generally from the medial side (between gracilis/semitendinosus and gastrocnemius). It is obvious that the authors did the same. I suggest that they add this to the text and add to the legend of Fig. 1A that this is a medial view.

4. Results, lines 201-204. From these lines I conclude that the authors struggled with unexpected calcifications when approaching the arteries in the bodies without CT scans. It would be helpful to explain the problems and add it to the discussion as well. As it stands now it is only mentioned in lines 333-334, as part of limitations.

5. Figures. The quality of Figs 1A-F is not sufficient: not sharp enough and too dark (1B is very dark). This may be due to my copy but I advise to give it attention.

Reviewer #2: The authors present a small series of cadaveric experiments and conclude that fresh frozen cadavers accurately mimic the clinical scenario for lower extremity intervention. THe paper is well written with only minimal changes needed.

1) How long after death were the fresh frozen cadavers used? Months?

2) Exactly how did you thaw the cadavers? Over what period of time?

3) Why did you use only 1 formalin-fixed sample? This severely limits the conclusions. I agree that n=5 for the fresh frozen is sufficient to conclude that those work, but you have not ruled out fixed cadavers as a mimic to the clinic. This is especially important because fixed cadavers may be more easily identified.

6. PLOS authors have the option to publish the peer review history of their article (what does this mean?). If published, this will include your full peer review and any attached files.

Reviewer #1: **Yes: **Ronald L.A.W. Bleys

Reviewer #2: No

---

## [Author Response · Author response to Decision Letter 0]

13 Apr 2023

We thank reviewers and the Academic Editor for their time and effort and appreciate the helpful comments and suggestions. All reviewer comments have been addressed and we believe this has considerably improved the manuscript. 

Specifically, we have adjusted the wording of the anatomical structures as well as added the requested additions to the material and methods section regarding storage and preparing of the human cadavers and surgically preparation. 

We have also made some minor changes to improve the clarity of the manuscript, which do not alter the content. 

Reviewer #1:

The authors have developed an extracorporeal perfusion model for the femoral artery in human cadavers. This model is of interest for research and training purposes, especially in the fields of radiology and vascular surgery. Therefore, the model may also contribute to the reduction of the use of large animals such as pigs, which are sometimes used for training purposes.

The authors have described setting up the model in great detail. This will certainly help anatomists and clinicians to set up a similar model and experiment with other applications as well.

The paper is well written and well discussed including its potential applications and limitations.

I have the following comments.

1. The authors have decided to use the common clinical terms for the arteries instead of the official anatomical terminology. This is fine but I suggest that they:

- explain the choice for terminology. In Terminologia Anatomica: femoral artery gives off the deep artery of the thigh (as a branch). Clinical terminology: common femoral artery divides into superficial and deep femoral arteries.

- replace “profound femoral artery” by “deep femoral artery”

Thank you for this comment. We have chosen to use the commonly used names of vessels in clinical parlance because the paper is aimed at a radiological or angiological readership. The difference in naming regarding "profound" and "deep" may be locally divergent. As recommended, "profound femoral artery" was changed to "deep femoral artery."

2. M&M, lines 37-39. Formalin is a solution of 37% formaldehyde. It is common to use a 4% formaldehyde solution for fixation purposes. This is approximately 10% formalin. It seems to me that 10% formaldehyde, as the authors state, is not correct.

Thanks for pointing out this inaccuracy. As you suspected, it is a 10-fold diluted formalin solution with a concentration of ~4 % formaldehyde. The information has been corrected accordingly in the manuscript.

3. M&M, line 50. The infragenicular approach of the popliteal artery is generally from the medial side (between gracilis/semitendinosus and gastrocnemius). It is obvious that the authors did the same. I suggest that they add this to the text and add to the legend of Fig. 1A that this is a medial view.

Thank you for this note. As suggested, the infragenicular access has been specified in the text and added to the legend of Figure 1.

4. Results, lines 201-204. From these lines I conclude that the authors struggled with unexpected calcifications when approaching the arteries in the bodies without CT scans. It would be helpful to explain the problems and add it to the discussion as well. As it stands now it is only mentioned in lines 333-334, as part of limitations.

This comment is reasonable. Vascular calcifications, especially in the inguinal access, as well as higher grade calcifications in the vascular course complicate the preparation and the establishment of perfusion. We have added a corresponding paragraph (results & discussion).

5. Figures. The quality of Figs 1A-F is not sufficient: not sharp enough and too dark (1B is very dark). This may be due to my copy but I advise to give it attention.

Thank you for this note. The poor quality of the images is most likely due to the compression during the creation of the PDF file. We have checked the original figures again. The figures meet the journal's specifications in terms of size and resolution. The desired structures are recognizable.

Reviewer #2: 

The authors present a small series of cadaveric experiments and conclude that fresh frozen cadavers accurately mimic the clinical scenario for lower extremity intervention. The paper is well written with only minimal changes needed.

1) How long after death were the fresh frozen cadavers used? Months?

The fresh-frozen cadavers were deep-frozen for only a short time of several weeks to a few months before they were used for preparation. They were frozen with the shortest possible latency after the death of the body donors (< 24h).

2) Exactly how did you thaw the cadavers? Over what period of time?

Thank you for the helpful feedback. The cadavers were stored in a -20°C freezer. For the thawing process, the cadavers were stored in a refrigerated room at about 7°C for 2 to 3 days depending on the body mass of the body donors. We have included this fact and the freezing time in the subsection "Human Cadavers".

3) Why did you use only 1 formalin-fixed sample? This severely limits the conclusions. I agree that n=5 for the fresh frozen is sufficient to conclude that those work, but you have not ruled out fixed cadavers as a mimic to the clinic. This is especially important because fixed cadavers may be more easily identified.

Thanks for the comment, the argument is reasonable. We would have preferred formalin-fixed bodies as well. On the one hand formalin-fixed body donors are no less rare than fresh-frozen cadavers. Available body donors are preserved either chemically or thermally, depending on requirements and suitability for further use. Moreover, formalin-fixed cadavers are preferably used for dissection courses and cannot/should not be damaged beforehand. After the courses, in contrast, they are no longer suitable for use as models, since the soft tissue is completely removed step by step. 

The joints as well as the soft tissues of the formalin-fixed cadavers are so rigid that the classic surgical accesses, both inguinal and infragenicular, cannot be used or lead to unacceptable soft tissue damage resulting in artifacts and leakages. This is a general fact and not an individual problem of the used formalin-fixed body. 

In summary, even with only one body donor, it can be shown that the perfusion model cannot be established in the manner presented if using formalin-fixed cadavers. For a transfer, the model would have to be modified to such an extent that it would become unsuitable for the intended purpose.

---

## [Decision Letter · Decision Letter 1]

2 May 2023

Continuous extracorporeal femoral perfusion model for intravascular ultrasound, computed tomography and digital subtraction angiography

PONE-D-23-03751R1

Dear Dr. Gruschwitz,

We’re pleased to inform you that your manuscript has been judged scientifically suitable for publication and will be formally accepted for publication once it meets all outstanding technical requirements.

Kind regards,

Peter R. Corridon

Academic Editor

PLOS ONE

Additional Editor Comments (optional):

Reviewers' comments:

Reviewer's Responses to Questions

**Comments to the Author**

1. If the authors have adequately addressed your comments raised in a previous round of review and you feel that this manuscript is now acceptable for publication, you may indicate that here to bypass the “Comments to the Author” section, enter your conflict of interest statement in the “Confidential to Editor” section, and submit your "Accept" recommendation.

Reviewer #1: All comments have been addressed

2. Is the manuscript technically sound, and do the data support the conclusions?

Reviewer #1: (No Response)

3. Has the statistical analysis been performed appropriately and rigorously? 

Reviewer #1: (No Response)

4. Have the authors made all data underlying the findings in their manuscript fully available?

Reviewer #1: (No Response)

5. Is the manuscript presented in an intelligible fashion and written in standard English?

Reviewer #1: (No Response)

6. Review Comments to the Author

Reviewer #1: (No Response)

7. PLOS authors have the option to publish the peer review history of their article (what does this mean?). If published, this will include your full peer review and any attached files.

Reviewer #1: **Yes: **Ronald L.A.W. Bleys

---

## [Editor Report · Acceptance letter]

15 May 2023

PONE-D-23-03751R1 

Continuous extracorporeal femoral perfusion model for intravascular ultrasound, computed tomography and digital subtraction angiography 

Dear Dr. Gruschwitz:

I'm pleased to inform you that your manuscript has been deemed suitable for publication in PLOS ONE. Congratulations! Your manuscript is now with our production department. 

Kind regards, 

on behalf of

Dr. Peter R. Corridon 

Academic Editor

PLOS ONE